# Effects of Beta-Blockers on Melanoma Microenvironment and Disease Survival in Human

**DOI:** 10.3390/cancers12051094

**Published:** 2020-04-28

**Authors:** Ludovic Jean Wrobel, Angèle Gayet-Ageron, Frédérique-Anne Le Gal

**Affiliations:** 1Hôpital Cantonal Universitaire de Genève, Service de Dermatologie, Rue Gabrielle Perret-Gentil 4, 1211 Genève (GE), Switzerland; 2Centre de Recherche Clinique & Division of Clinical-Epidemiology, Department of Health and Community Medicine, University of Geneva & University Hospitals of Geneva, 1205 Genève (GE), Switzerland

**Keywords:** melanoma, beta-blocker, anti-tumor immune response, survival

## Abstract

*Background:* The regulation of melanoma by noradrenergic signaling has gain attention since pre-clinical and clinical studies suggested a benefit of using beta-blockers to control disease progression. We need to confirm that human melanoma recapitulates the mechanisms described from pre-clinical models. *Methods:* The sources and targets of norepinephrine in the microenvironment of 20 human melanoma samples was investigated using immunostaining. The effect of an exposure to beta-blockers on immune cell type distribution and expression of immune response markers was assessed with immunostaining on 212 human primary melanoma. A statistical analysis explored the effect of an exposure to beta-blockers on progression free survival, melanoma related survival, and overall survival on the 286 eligible patients. *Results:* Tumor cells and macrophages may be a source of norepinephrine in melanoma microenvironment. Tumors from patients exposed to wide spectrum beta-blockers recapitulate the increased infiltration of T-lymphocytes and the increased production of granzyme B observed in pre-clinical models. An exposure to beta-blockers is associated with a better outcome in our cohort of melanoma patients. *Conclusion:* This study shows the association between an exposure to wide spectrum beta-blockers and markers of an effective anti-tumor immune response as well as the protective effect of beta-blockers in human melanoma patients.

## 1. Introduction

There is growing evidence that the catecholamine norepinephrine (NE) plays a role in melanoma progression and the associated immune suppression. One of the hypotheses is that the sympathetic nervous system releases norepinephrine in lymphoid organs and at the tumor site promoting both tumor progression and immune suppression [1,2]. In vitro, norepinephrine can modulate initiative, proliferative, and effector phases of the immune response by sculpting cytokine production profile and regulating immune cell populations’ proliferation [1]. In a murine model of melanoma, a daily treatment with the wide spectrum beta-blocker propranolol decreases the fraction of myeloid suppressor cells and increases the fraction of cytotoxic lymphoid cells [3] infiltrating the tumor. This strengthens the idea that noradrenergic signaling regulates anti-tumor immune response. Besides, melanocytes and keratinocytes have been shown to express the different enzymes for norepinephrine synthesis [4,5] suggesting a possible autocrine or paracrine noradrenergic stimulation on tumor cells. In vitro, melanoma cell lines produce pro-angiogenic and pro-tumor factors upon noradrenergic stimulation [2] suggesting a direct involvement of noradrenergic signaling in the regulation of tumor cells and disease extension.

The sources of norepinephrine, the different receptors involved as well as the local versus systemic regulations induced by catecholamines in human melanomas is still a matter of debate [1,6,7,8]. In the present study we investigated the expression of norepinephrine and its target receptors in human melanoma tumors.

Using a cohort of melanoma patients from our archived tissue bank, we assessed the effect of the exposure to various types of beta-blockers on the local immune pattern and melanoma progression. We completed our study by a survival analysis assessing the association between exposure to beta-blockers and melanoma related death or disease progression. Our findings in melanoma patients who are on a beta-blocker treatment largely recapitulate our previous observations in animal models.

## 2. Results

### 2.1. Adrenergic Sources and Targets in Melanoma Tumors

We investigated the presence of noradrenergic sources at the tumor site. The common hypothesis is that nerve fibers, coming from the peripheral nervous system innervating the skin, brings a noradrenergic input that triggers tumor development. Comparing human samples from normal skin with nevi (benign and dysplastic) and melanoma, we observed a significantly higher innervation of all melanocytic lesions (Figure 1A). These fibers likely originating from the subepidermal nerve plexus were negative for tyrosine hydroxylase (TH) and norepinephrine (NE, Figure 1B,C).

On a series of 20 human melanomas, we used immunohistochemistry to assess the expression of norepinephrine in the tumor tissue. Norepinephrine was only detected in tumor cells and macrophages (Figure 2A,B, Table 1). In macrophages, norepinephrine is mainly detected in intracellular granules. In melanoma cells, we observed two different patterns coexisting in the tumors. Some norepinephrine was present at the surface of melanoma cells’ membrane, likely to represent receptor-bound norepinephrine. We also observed norepinephrine co-localizing with the cytoplasm of melanoma cells. Both macrophages and melanoma cells expressed the norepinephrine synthesizing enzyme dopamine beta-hydroxylase (DBH) (Figure 2C,D).

In order to identify putative targets for beta-blockers, we assessed the expression of the two main adrenergic receptors ADRB1 and ADRB2. We observed the expression of ADRB1 receptors on macrophages and on mast cells (Figure 3A,B). Melanoma cells, keratinocytes and a subset of T-cells were positive for ADRB2 (Figure 3C–E, Table 1).

Since some beta-blockers may interact with serotonergic signaling [9,10,11], we controlled for the expression of serotonin, 5HT1A and 5HT1B receptors. We did not detect any expression of serotonin (5HT) in our melanoma samples. In contrast, tumor cells, keratinocytes and T-cells were positive for 5HT1A, while only blood vessels were positive for 5HT1B receptor (Figure 4, Table 1).

These results indicate that norepinephrine is produced in the tumor microenvironment and that targets for beta-blockers are expressed in situ.

### 2.2. Patient Characteristics

286 patients diagnosed as having primary melanoma were included in the study between 1 January 1987 and 29 April 2014 at the Department of Dermatology of the University Hospitals of Geneva, Switzerland. We limited our selection to patients diagnosed with melanoma at least 5 years ago. We therefore benefit from a significant follow-up period to assess the effect of an exposure to beta-blockers on the risk of death or disease progression. Selection strategy is presented in the Figure 5. 

We distinguished three study groups based on their exposure to beta-blockers. Table 2 gives an overview of patients’ characteristics in each study group (*n* = 230 in the control group, *n* = 41 in the cardioselective beta-blockers group and *n* = 15 in the wide-spectrum beta-blockers group). The two groups of patients exposed to beta-blockers at melanoma diagnosis were significantly older than patients of the never exposed group. We did not find any differences among groups regarding other characteristics or clinical variables.

### 2.3. Exposure to Beta-Blockers and Histopathology of Melanoma

Table 3 compares the histologic characteristics of melanoma tumors by groups of exposure to beta-blockers. We observed immune cell populations distribution in tumor environment as well as the expression of key factors regulating disease progression. In total 1.9% to 30.7% of the values were missing due to the limited access to tumor material in order to preserve patient samples, as stated in our ethic agreement. 

Patients incidentally exposed to any type of beta-blocker (i.e., cardioselective beta-blocker or wide spectrum beta-blocker) had a significantly lower intra-tumor blood vessel density, as assessed by CD34 staining. Lymph vessel density assessed by D2-40 (podoplanin) staining was not significantly different across the three groups of exposure to beta-blockers (Table 3). Only patients exposed to wide spectrum beta-blockers show a significantly lower Ki67 index indicating a lower proliferation rate in the tumors of that group of patients. In the same way, only the tumors of patients exposed to wide spectrum beta-blockers exhibit a higher CD3 + T-cell density and a significantly higher granzyme B expression (Table 3). Exposure to beta-blockers was not associated with a significant difference in the distribution of macrophages and neutrophils infiltrating the tumor (Table 3). Of note, we distinguished between two major patterns of infiltration by neutrophils. These two patterns are present in our three study groups. In most of the samples (85%; *n* = 144) infiltration by neutrophils was scarce but a subset of the samples in each study group show a large neutrophil infiltrate (more than 50 cells per mm^2^).

We assessed the presence of mast cells by staining for mast cell tryptase antigen. The exposure to cardio-selective beta-blockers only is correlated with a higher density of mast cells in the tumor microenvironment. We did not observe any significant difference in MHC class II molecules expression between each study group.

The expression of TNF alpha was not significantly different in our three study groups despite a trend towards a higher production of TNF alpha in the tumors of patients exposed to wide spectrum beta-blockers. Likewise, the expression of inducible nitric oxide synthase (iNos) was not significantly different among the three study groups despite a trend towards a lower production of iNos in the tumors of patients exposed to wide spectrum beta-blockers.

Finally, we observed a higher density of cells expressing IL-10 in samples from patients exposed to wide spectrum beta-blockers but not cardio-selective type.

### 2.4. Survival Analyses

Only one patient exposed to wide spectrum beta-blocker experienced a progression event and none of them died from melanoma related death, this result in a lack of statistical power due to the lack of event. We pooled cardioselective and wide spectrum beta-blocker into one exposed group. We considered two types of exposure to beta-blockers: (1) exposure before the diagnosis of melanoma (no exposure vs. exposure with cardioselective or non-cardioselective beta-blockers), and (2) exposure to beta-blockers before the diagnosis of melanoma and during follow-up (never users vs. users before the diagnosis of melanoma vs. users after the diagnosis of melanoma). Among 229 patients never exposed to beta-blockers before diagnosis, 24 patients experienced an exposure to beta-blockers after the diagnosis of melanoma (21 under cardioselective and 3 under wide spectrum beta-blocker therapy). In this group exposed to beta-blockers after primary melanoma resection, the exposure to beta-blockers began 47.2 +/− 59.5 months (mean +/− SD, range 8–288 months) after the resection of the primary tumor. These patients secondarily exposed to beta-blockers were significantly older (mean age 70.4 ± 11.3 years) than patients in the control group. Our primary endpoints were progression free survival and melanoma related survival. Table 4 shows the analyses of the effect of beta-blocker exposure on overall survival, melanoma related death and progression free survival. Age was initially used as a binary variable because it does not fully respect the assumption of Cox proportional hazard model where continuous variables should be log-linearly associated with the outcome. This assumption was acceptable for overall survival model, so we provide the results using age as a continuous variable in this model only.

#### 2.4.1. Progression-Free Survival

A greater Breslow index and ulceration were associated with a higher risk of disease progression in univariate and multivariable model. The use of beta-blockers was significantly associated with progression-free survival in univariate analyses. The use of beta-blockers before diagnosis and during follow up was significantly associated with a lower risk of disease progression; there was a difference in the hazard of disease progression between beta-blocker users depending on the time of introduction of the beta-blocker (before or after diagnosis). In multivariable analyses, the overall use of beta-blockers was not significantly associated with disease progression (*p* = 0.086), but the group of patients exposed to beta-blockers before diagnosis and during follow up has a significantly lower hazard of disease progression compared to non-exposed patients (*p* = 0.042). 

#### 2.4.2. Melanoma Related Survival

The risk of melanoma related death was significantly increased with a greater Breslow index and ulceration of the tumor in univariate analyses. The use of beta-blockers was significantly associated with a lower hazard of melanoma related death (*p* = 0.034), in particular the hazard was lower among patients exposed to beta-blockers before diagnosis as compared to non-exposed patients (*p* = 0.023). In multivariable analyses, only Breslow index remains significantly associated with a higher risk of melanoma related mortality.

#### 2.4.3. Overall Survival

The use of beta-blockers is not significantly associated with all-cause mortality but the hazard of all-cause death was significantly lower in patients exposed to beta-blockers after the diagnosis of melanoma compared to never exposed patients (Table 4), or compared to patients exposed to beta-blockers before the diagnosis of melanoma (HR = 0.20; 95%CI: 0.05-0.92, *p* = 0.039). The hazard of all-cause mortality was also significantly and independently associated with a greater Breslow index and age.

## 3. Discussion

The repurposing of beta-blockers as anti-tumor agents is being considered in several cancers. On the one hand, pre-clinical studies showed that beta-blockers can block cell proliferation, migration, invasion, resistance to therapy, and metastasis [3,8,12]. On the other hand, there is an accumulation of data showing the link between noradrenergic stimulation and pro-tumor processes [2,13]. Recently, a first prospective study highlighted the protective effect of an adjuvant treatment with the wide spectrum beta-blocker propranolol against melanoma related death [14] confirming the previous observation in retrospective cohorts [15,16,17]. Despite the limitations of the trial, related to the small sample size and the short follow-up, the authors observed a significant protective effect of propranolol against the risk of disease recurrence and metastases development. 

We first investigated whether the effect of beta-blockers rely on systemic regulations only, or can be partially attributed to local actions. We showed that norepinephrine and the main adrenoceptors ADRB1 and ADRB2 are present in the tumor microenvironment. Despite the hypothesis that norepinephrine is delivered locally by sympathetic innervation, we observed that nerve fibers surrounding the tumors are negative for norepinephrine and catecholamines synthesizing enzymes. The increased nerve fiber density around all melanocyte lesions may be related to the Schwann cell origin of melanocytes. Schwann cells are necessary for peripheral nerve survival and melanocytes like Schwann cells may release factors increasing these fibers survival in the microenvironment of melanocyte lesions [18] In contrast, we showed that macrophages carry granules containing norepinephrine and may be a source of noradrenergic supply in the tumor microenvironment (Figure 2B). Tumor cells exhibited two different patterns of norepinephrine staining. We detected norepinephrine in the cytoplasm of melanoma cells, suggesting production of norepinephrine and/or internalization. The expression of dopamine beta-hydroxylase suggests that tumor cells may have the ability to produce norepinephrine and may be another source of norepinephrine in the tumor environment. Some tumor cells also displayed a membrane staining likely to represent membrane bound norepinephrine. In vitro studies have highlighted the potential of melanoma cells to produce pro-tumorigenic factors upon noradrenergic stimulation [2], mainly through an ADRB2 stimulation. ADRB2 expression was detected on keratinocytes, tumor cells and a subset of T-cells. We can hypothesize than norepinephrine released by macrophages and/or tumor cells participates to the establishment of a pro-tumorigenic microenvironment.

Interestingly, we observed the expression of ADRB1 on macrophages and mast cells, two cell types known to predominantly express ADRB2 in the literature. Both were negative for ADRB2, which suggests that macrophages and mast cells could be of a particular phenotype in the microenvironment of melanoma. 

Altogether, these results show that norepinephrine and its receptors are present locally in the tumor and its microenvironment. The wide expression of beta-adrenoceptors in the tumor environment and the presence of norepinephrine provide local targets for beta-blockers. Affecting tumor cells, immune cells and keratinocytes, noradrenergic signaling may rule different aspects of tumor progression. 

We next assessed the expression of different biomarkers of melanoma progression on a cohort of patients exposed or not exposed to beta-blockers at the time of diagnosis. We distinguished the exposure to cardio-selective beta-blockers from the exposure to wide spectrum beta-blockers. Despite the small size of our sample, we could observe some significant differences between our three study groups. We have to note that among our patients exposed to wide spectrum beta-blockers, 9 were treated with eye drops and 6 per os without any major difference in the histopathological data. Eye drops of beta-blockers are known to induce systemic effects [19,20] and our results suggest that the administration route does not affect the protective effects of beta-blockers observed on human melanoma tumors. Further studies should investigate the optimal dose of beta-blockers as an adjuvant treatment for melanoma and assess the effect of the administration route on reaching that optimal dose.

Pre-clinical studies in animal models of melanoma suggested that exposure to the wide spectrum beta-blocker propranolol results in an increased infiltration of T-lymphocytes and an increased cytotoxic activity [3]. This increased anti-tumor immune response was associated to a reduced infiltration of the tumor by macrophages and polymorphonuclear neutrophils [3]. We hypothesized that propranolol affected T-lymphocytes indirectly, by reducing immune suppression induced by macrophages and polymorphonuclear neutrophils. Here we observed a significantly higher number of CD3+ T-lymphocytes in the tumor of patients exposed to wide spectrum beta-blockers only. In this study group, T-cells exhibited an increased expression of Granzyme B suggesting a higher cytotoxic potential. In contrast, we found no significant difference in the number of CD68+ macrophages and myeloperoxidase+ neutrophils infiltrating the tumor. We cannot exclude a switch in the immune suppressive activity of these myeloid cells when exposed to wide spectrum beta-blockers, but this result suggests that the stimulation of cytotoxic T-cell activity may not result from a reduced infiltration by myeloid suppressor cells. Nevertheless, the way that wide spectrum beta-blockers may release the activity of T-cells and increase their infiltration of the tumor in humans has to be further investigated. We have to note the significantly higher amount of cells expressing IL10 in the tumor stroma of patients exposed to wide spectrum beta-blockers. Despite being historically associated with immune suppression, different studies showed a protective effect of IL10 against tumor progression [21,22]. The role of IL10 on the local anti-tumor response remains to be elucidated.

Since the observation of the dramatic effect of propranolol on the involution of infantile hemangiomas [23,24], beta-blockers are considered for their putative anti-angiogenic properties. In our previous studies on models of melanoma we showed that a daily treatment with the wide spectrum beta-blocker propranolol induces not only a decrease in tumor blood vessel density but also a reduced tumor cells proliferation index, i.e., Ki67 index [12]. In patients exposed to wide spectrum beta-blockers, we observed a significantly lower proliferation index (i.e., Ki67 index) than in the cardioselective beta-blocker or the unexposed study groups. This observation strengthens the hypothesis of an inhibition of melanoma cells proliferation by wide spectrum beta-blockers but not by cardioselective beta-blockers. In contrast, the exposure to both types of beta-blockers is associated with a lower blood vessel density inside the tumor. Confirming our previous analyses on animal models of melanoma, these observations suggest that cardio-selective and wide spectrum beta-blockers affect tumor angiogenesis during melanoma progression.

Furthermore, we observed that the exposure to both types of beta-blockers is associated to a reduced density of CD34+ stromal fibroblasts (vimentine +). These cells are suggested to be involved in angiogenesis, tumor stroma formation, matrix remodeling and immunosuppression [25]. Beta-blockers, by decreasing the density of CD34+ fibroblasts may impair melanoma growth and slow down disease progression.

The role of mast cells in cancer initiation and progression is highly debated [26,27,28,29]. In melanoma, mast cells are suggested to have pro-tumor and pro-angiogenesis properties [28,29], but other elements suggest a more anti-tumor phenotype depending on the mediators secreted [27,28]. The higher infiltration of mast cells in the tumors of patients exposed to cardioselective beta-blockers is difficult to explain with a histopathological study and this phenomenon has to be further investigated. Nevertheless, care should be taken with the use of cardioselective beta-blockers for patients suffering from melanoma since the role of mast cells in cancer progression is not yet understood.

Altogether these results suggest that wide spectrum beta-blockers are promising agents in adjuvant therapy for melanoma. Only wide spectrum beta-blockers are associated with biomarkers of an increased anti-tumor immune response and a decreased tumor growth. Nevertheless, cardio-selective beta-blockers may affect partially tumor progression modulating tumor angiogenesis and CD34+ stromal fibroblast density. 

We have to note that among patients exposed to wide spectrum beta-blockers (*n* = 15) only one patient experienced a progression event and none of them died from melanoma. In light of this limitation and the lack of power related to the low number of events we pooled both type of beta-blockers into one group for survival analyses. Our results indicate that exposure to beta-blockers before melanoma diagnosis and during the follow up period significantly decreases the risk of disease progression.

We should be careful in concluding on the apparent protective effect of beta-blockers on melanoma related survival. Given the large period covered by our study, patients who developed metastases underwent different medical strategies of metastasis management due to the evolution of the guidelines over time and the development of new therapies. This interacts with the measured outcome and represents a potential confounding factor in the analysis of the effect of beta-blockers. Unfortunately, given the variety of therapies and the small size of the population exposed to beta-blockers and experiencing metastases, we cannot evaluate this interaction with sufficient power in our cohort. Further studies are needed to understand the possible interactions between the exposure to beta-blockers and the medical management of metastasis, especially since beta-blockers modulate anti-tumor immune response and may enhance or impair the efficacy of immune checkpoint inhibitors, as well as modulating resistance to treatment.

Despite a trend towards a better outcome for patients exposed to beta-blockers after the resection of the primary tumor, the effect of a delayed introduction on melanoma specific survival or progression free survival was not significant. This is probably related to the small sample size, the long delay before beta-blocker introduction, as well as the low fraction of patients exposed to wide spectrum beta-blockers in this particular group.

## 4. Materials and Methods

The investigations were carried out following the rules of the Declaration of Helsinki of 1975, revised in 2013. The collection of material and the analyses performed were allowed by our local ethic committee (Commission cantonale d’éthique de la recherche de Genève) and were registered under project n° 15-092.

### 4.1. Study Design

Retrospective study of patients diagnosed with a first primary melanoma, in a single center, with a clinical record for at least five years from primary melanoma diagnosis.

### 4.2. Study Setting

286 patients diagnosed as having primary melanoma were included in the study between 1 January 1987 and 29 April 2014 at the Department of Dermatology of the University Hospitals of Geneva, Switzerland. A subset of 212 patients was considered for immuno-histochemical analyses, when the primary tumor sample was available and the remaining tumor tissue reliable with the original recorded Breslow index. Our primary endpoints were the effect of an exposure to beta-blockers on progression free survival and melanoma related death. The effect of beta-blocker exposure on overall survival was assessed as a secondary outcome.

### 4.3. Selection Strategy

The original records and the selection strategy are presented in Figure 5. Patients were considered eligible when presenting a first primary melanoma (in situ melanoma excluded according to the low risk of progression), with reliable clinical data related to beta-blocker exposure. Only patients for whom electronic clinical record, follow up interviews and referring family physician inputs could confirm or exclude the use of beta-blockers and confirm compliance to treatment were included. When the information was incomplete or doubtful the patient was excluded from the study. 

### 4.4. Sample Preparation

Tissue blocks were sectioned in 7 µm slices using a standard microtome and mounted on superfrost plus slides (Menzel-Gläser, Braunschweig, Germany). The tissue sections were oven-dried overnight at 52 °C. Before starting staining protocols, the slides were dewaxed in four baths of Ultraclear (VWR, Datmstadt, Germany) for 5 min each. The slides were rinsed in four baths of absolute ethanol for 3 min each and kept in distilled water until use.

### 4.5. Heat Induced Epitope Retrieval

Prior to all immunostainings presented in this manuscript, an epitope recovery using citrate buffer pH6 was performed. Citrate buffer was prepared by diluting 1.92 g of anhydrous citric acid (Sigmaaldrich, Datmstadt, Germany) in 1 L of distilled water. The acidity was adjusted to pH6 with 5N NaOH under constant monitoring with a pHmeter (Metrhom, Herisau, Switzerland). The citrate buffer solution was then warmed to boiling point using a standard microwave oven at power 900 Watts. The tissue slides were dived in the boiling buffer and kept in microwave oven for 16 min at 350 Watts. The bucket containing the citrate buffer and the slides was allowed to cool down at room temperature for 30 min. The slides were then rinsed in distilled water 2 times for 1 min and transferred to phosphate buffer saline (PBS, Gibco, Dreieich, Germany) for further staining procedures.

### 4.6. Staining Procedure

Primary antibodies (see Appendix A for species, specificity and dilution factors) were diluted in antibody diluent (ab64211, Abcam, Cambridge, UK) at the desired concentration. The tissue sections were incubated in the primary antibody mix for 2 h. The slides were rinsed in 3 baths of PBS for 5 min each before incubation with the secondary antibody mix. Secondary antibody mix was obtained by diluting the desired secondary antibodies in antibody diluent (Appendix A for species, specificity and dilution factors). The slides were rinsed in 2 baths of PBS for 5 min each and mounted under coverslips with mounting medium containing DAPI (Fluoromount-G, ThermoFisher, Dreieich, Germany).

### 4.7. Image Acquisition and Analyses

The acquisition of images for qualitative analyses was performed on a Zeiss LSM800 airyscan confocal microscope (Zeiss, Jena, Germany). The quantification of cell populations and the quantification of markers’ expression was performed on wide pictures acquired with an axioscan Z1 slide scanner (Zeiss, Jena, Germany). The quantification of markers and cell population was obtained with Definiens Tissue studio software (Definiens AG, Munich, Germany). The quantification of nerve fiber density was done on ImageJ using Feature J hessian plugin.

### 4.8. Quality Control for Image Quantification

Automated image analyses need quality control steps to guarantee the reliability and accuracy of the estimation based on image processing. This is particularly true when a large amount of pictures from different datasets (or experiment replications) are compared. Our preliminary work consisted in creating quantification routines to establish guidelines for batch analyses and standardize the work of the experimenter. Figure 6 summarizes the workflow we follow for image processing to control the quality of the results.

### 4.9. Drugs

Three subgroups of patients have been exposed to different types of beta-blockers before melanoma diagnosis and during the follow-up period. Cardio selective beta-blockers are selective antagonists for beta-1 adrenoceptors as opposed to wide spectrum beta-blockers which are non-selective antagonists of beta-1 and beta-2 adrenoceptors. Table 5 summarizes the different molecules and their proportion in each subgroup.

### 4.10. Statistics

#### 4.10.1. Histopathological Data

We distinguished three study groups based on their exposure to beta-blockers: a control group consisted in patients who were never exposed to any type of beta-blockers at diagnosis of melanoma, and two groups of patients exposed to cardioselective beta-blockers or wide spectrum beta-blockers respectively. We used Chi-2 or Fischer’s exact tests for the comparisons of categorical variables by groups of exposure to beta-blockers, and Kruskal–Wallis nonparametric test for comparisons of continuous variables (by groups of exposure to beta-blockers). For the comparisons of histological characteristics, we performed pair-wise post-hoc comparisons applying the Nemenyi test. 

#### 4.10.2. Survival Analyses

Competing risk models were performed to assess the association between exposure to beta-clockers and progression free survival, melanoma related survival or overall survival. We performed three different survival analyses: (1) Progression free survival (without clinical progression or melanoma-related mortality). The follow-up started on the date of melanoma diagnosis and ended on the date of the first event of recurrence, death for any cause or on the date of last information (i.e., censored observations); (2) melanoma-related survival where we considered other cause of death or death of unknown cause as competing risk events; and (3) overall survival where the follow-up started on the date of melanoma diagnosis and ended on the date of death for any cause or on the date of last information (i.e., censored observations). A Cox proportional hazards model was used to evaluate the effect of beta-blocker treatment on overall mortality, independently of other significant prognostic factors. Competing risks Cox regression models were performed using other cause of death or death of unknown cause as competing risk events to evaluate the effect of beta-blocker treatment on melanoma-related mortality then on disease-free progression, independently of the same significant prognostic factors. The potential prognostic factors tested were: Breslow thickness, age (< vs. ≥ 60 years), and ulceration. All tests were two-tailed and *p* values < 0.05 were regarded as significant. We used STATA (version intercooled 16.0) for all analyses.

### 4.11. Data Sharing

All raw data for histopathological and survival analysis are available as Appendix A.

## 5. Conclusions

Taken together, the apparent cytotoxic switch of the local anti-tumor immune response in the wide spectrum beta-blockers exposed group and the better outcome of patients exposed to beta-blockers suggest that wide spectrum beta-blockers are promising agents to prevent disease progression. This highlights the need for wide scale prospective studies investigating the efficacy of wide spectrum beta-blockers as an adjuvant treatment for melanoma.

## Figures and Tables

**Figure 1 cancers-12-01094-f001:**
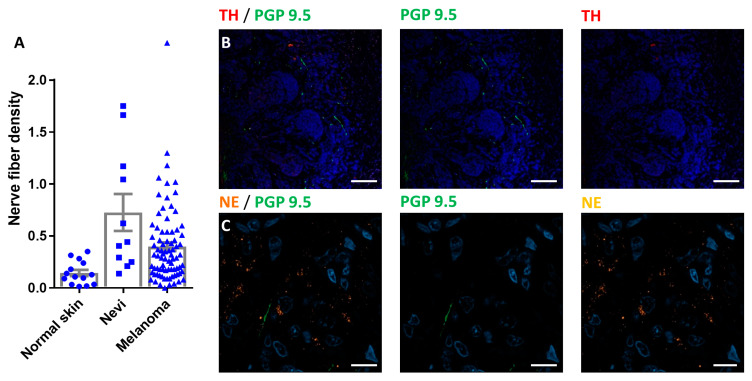
Subepidermal nerve plexus in melanoma tumors: (**A**) Histogram comparing the nerve fiber density under normal epidermis (normal skin), under nevi lesion (Nevi) or under melanoma tumor nodules (Melanoma). Results are given as mean +/− SEM. (**B**) Photographs of melanoma nodules surrounded by PGP 9.5 positive nerve fibers (green). These fibers are negative for tyrosine hydroxylase (TH, red). Scale bar 100 µm. (**C**) High magnification photograph showing norepinephrine positive melanoma cells (NE, orange) with PGP 9.5 positive / NE negative nerve fibers (green). Scale bar 10 µm.

**Figure 2 cancers-12-01094-f002:**
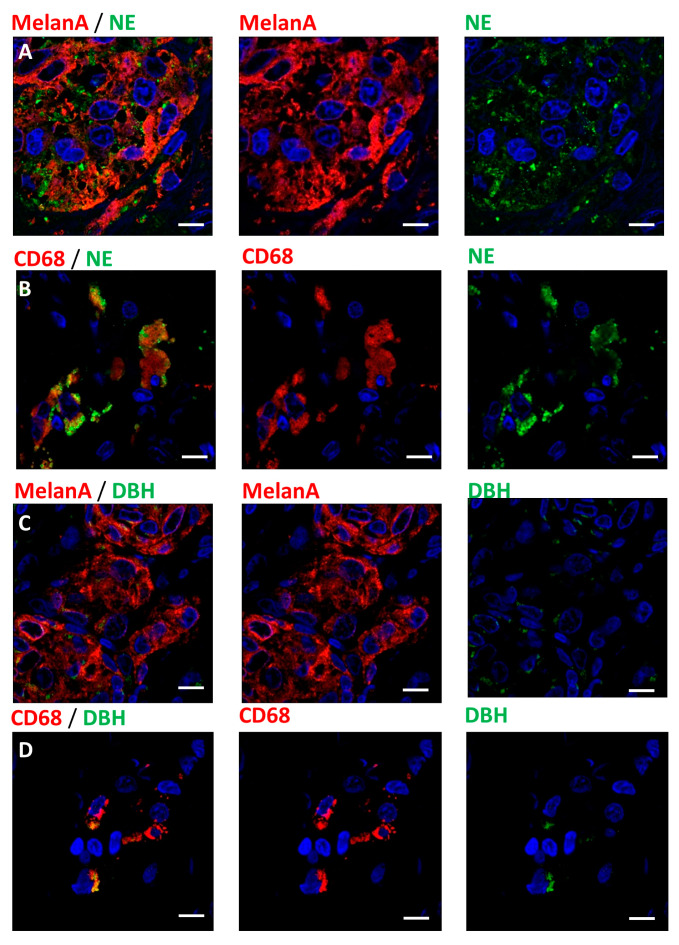
Norepinephrine expression in melanoma tumors: (**A**) Photographs showing melanoma cells stained for MelanA (red) antigen expressing norepinephrine (NE, green). (**B**) Photograph showing CD68 positive macrophages (red) carrying norepinephrine granules (NE, Green). (**C**) Photographs showing the expression of dopamine beta-hydroxylase (DBH, green) by melanoma cells (MelanA, red). (**D**) Expression of DBH (green) in a population of macrophages (CD68, red). Scale bars 10 µm.

**Figure 3 cancers-12-01094-f003:**
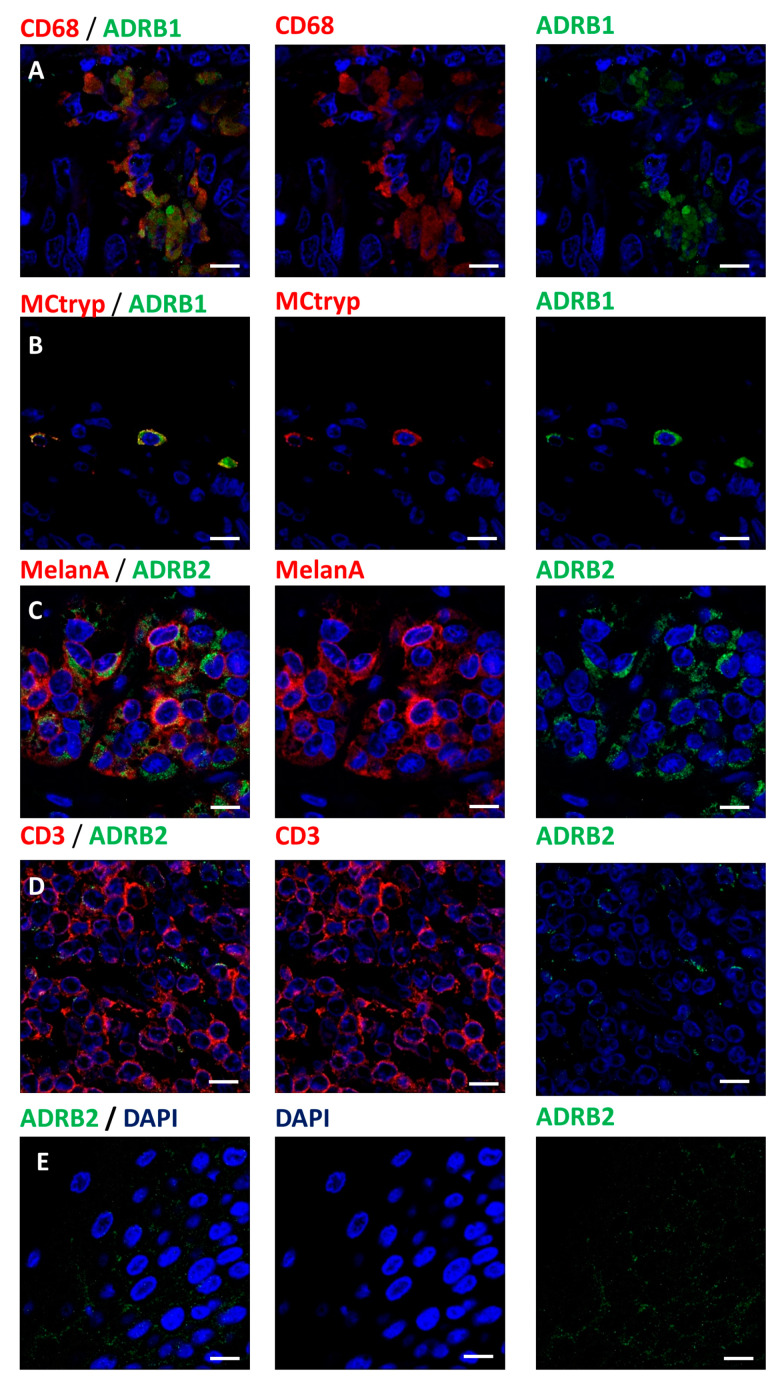
Expression of beta-adrenoceptors in melanoma microenvironment: (**A**) Expression of ADRB1 (green) receptor in CD68 (red) positive macrophages. (**B**) Tryptase positive mast cell (MCtryp, red) expression of ADRB1 (green) receptor. (**C**) Expression of ADRB2 (green) receptor in melanoma cells (MelanA, red). (**D**) T-cells (CD3, red) expression of ADRB2 (green) receptor. (**E**) Expression of ADRB2 (green) receptor in epidermal keratinocytes. Nuclei are counterstained with DAPI (blue) in every conditions. Scale bars 10 µm.

**Figure 4 cancers-12-01094-f004:**
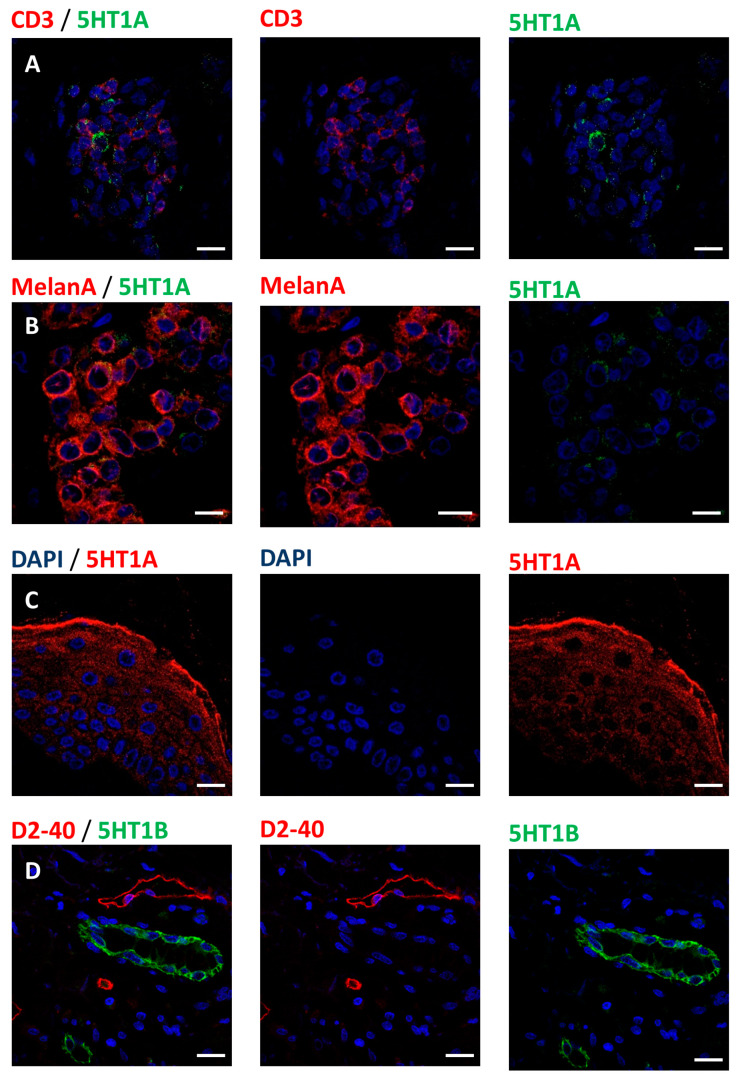
Expression of 5HT1A and 5HT1B serotonergic receptors in the microenvironment of melanoma tumors: (**A**) Expression of 5HT1A (green) receptor in a subpopulation of T-lymphocytes (CD3, red). (**B**) Melanoma cell (MelanA, red) expression of 5HT1A (green) receptor. (**C**) Section of epidermis showing the expression of the receptor 5HT1A (red) in epidermal keratinocytes. (**D**) Expression of 5HT1B (green) receptor in blood vessels, lymph vessels are stained for D2-40 (red) and do not express 5HT1B. Scale bars 10 µm.

**Figure 5 cancers-12-01094-f005:**
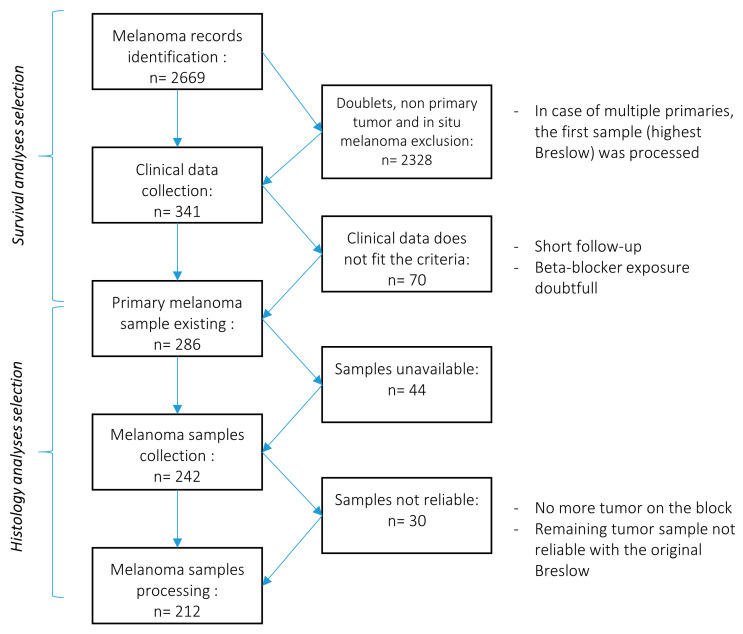
Selection strategy of for patients and samples validation: Diagram illustrating the selection process for the identification of eligible patients and samples.

**Figure 6 cancers-12-01094-f006:**
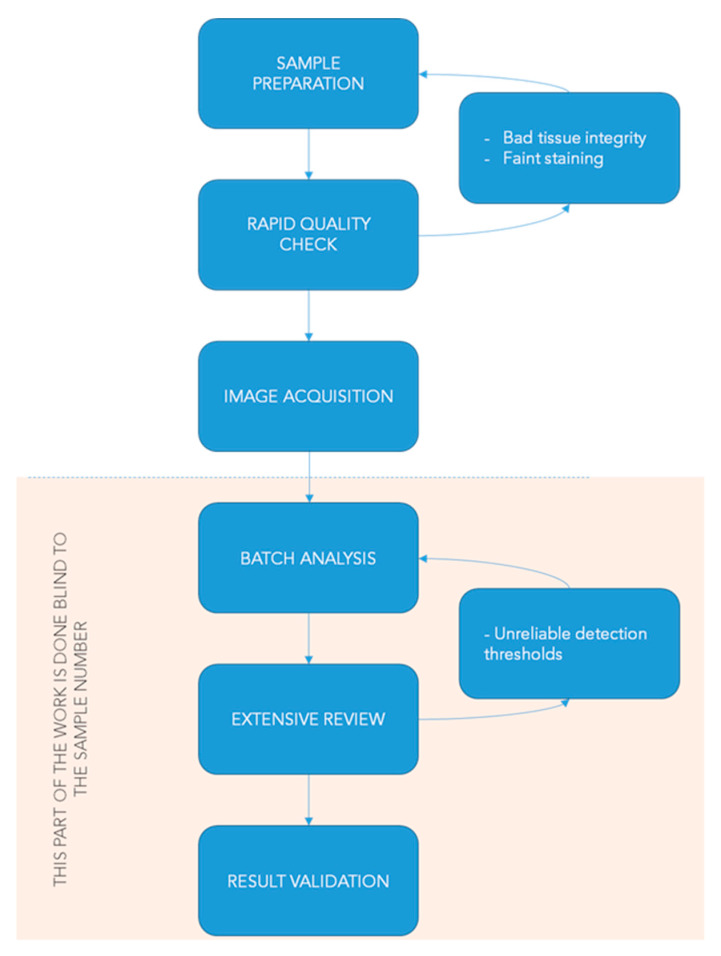
Workflow for automated image quantification and quality control.

**Table 1 cancers-12-01094-t001:** Expression of norepinephrine and targets for beta-blockers in melanoma tumor sections.

Cell Type	DβH	NE	ADRB1	ADRB2	5HT	5HT1A	5HT1B
Tumor cell	+	+	ND	+	ND	+	ND
Keratinocyte	+	ND	ND	+	ND	+	ND
Macrophage	+	+	+	ND	ND	ND	ND
Mast cell	ND*	ND	+	ND	ND	ND	ND
T-Cell	ND	ND	ND	+	ND	+	ND
blood vessels	ND	ND	ND	ND	ND	ND	+

* ND: not detected.

**Table 2 cancers-12-01094-t002:** Patient characteristics and clinical characteristics by treatment group (no exposure and exposure to beta-blockers) at diagnosis.

Variables	No Beta-Blockers (*n* = 229)	Cardioselective Beta-Blockers (*n* = 42)	Wide Spectrum Beta-Blockers (*n* = 15)	*p*-Value
Gender, *n* (%)				0.411
Female	96 (41.9)	13 (31.0)	6 (40.0)	
Male	133 (58.1)	29 (69.0)	9 (60.0)	
Age in years at diagnosis, mean (±SD, median)	59.4 (16.4, 62)	71.7 (10.6, 71.5)	69.3 (17.3, 70)	0.0001 ^*^
Mean Breslow thickness index (±SD, median)	1.66 (2.33, 0.80)	1.37 (1.82, 0.63)	1.51 (1.38, 1.50)	0.399 ^*^
Type of melanoma, *n* (%)				0.645
SSM	63 (27.5)	10 (23.8)	2 (13.3)	
NM	140 (61.1)	28 (66.7)	10 (66.7)	
Other	26 (11.4)	4 (9.5)	3 (20.0)	
Ulceration, *n* (%)	26 (11.4)	5 (11.9)	4 (26.7)	0.214 ^**^
Localization, *n* (%)				0.053 ^**^
Trunk	112 (48.9)	15 (35.7)	5 (33.3)	
Lower limb	65 (28.4)	9 (21.4)	3 (20.0)	
Upper limb	27 (11.8)	7 (16.7)	4 (26.7)	
Head & neck	25 (10.9)	11 (26.2)	3 (20.0)	
Clark level, *n* (%)				0.379 ^**^
II	80 (34.9)	17 (40.5)	7 (46.7)	
III	94 (41.1)	15 (35.7)	2 (13.3)	
IV	38 (16.6)	9 (21.4)	5 (33.3)	
V	14 (6.1)	1 (2.4)	1 (0.7)	
NA	3 (1.3)	0 (0)	0 (0)	
AJCC staging, *n* (%)				0.218 ^**^
IA	137 (60.1)	28 (66.7)	7 (46.7)	
IB	30 (13.2)	3 (7.1)	1 (6.7)	
IIA	13 (5.7)	5 (11.9)	4 (26.7)	
IIB	10 (4.4)	1 (2.4)	2 (13.3)	
IIC	6 (2.6)	1 (2.4)	0 (0)	
IIIA	12 (5.3)	0 (0)	0 (0)	
IIIB	5 (2.2)	2 (4.8)	1 (6.7)	
IIIC	10 (4.4)	1 (2.4)	0 (0)	
IIID	3 (1.3)	0 (0)	0 (0)	
IV	2 (0.9)	1 (2.4)	0 (0)	

* Comparisons were performed using Mann–Whitney non-parametric tests for continuous variables and Chi-2 test for categorical variables. ** Fischer’s exact test.

**Table 3 cancers-12-01094-t003:** Immuno-histologic characteristics at the time of diagnosis.

Variables	Overall	No Beta-Blockers	Cardio-Selective Beta-Blocker Users	Wide Spectrum Beta-Blocker Users	*p*-Value *	*p*-Value **	*p*-Value **
Intra tumor vessel density (*n* = 184)	1.33 (±1.31, 1.04, 0.57–1.64)	1.47 (±1.35, 1.17, 0.63–1.87)	0.99 (±1.23, 0.68, 0.30–1.31)	0.75 (±0.45, 0.63, 0.36–1.06)	0.001	0.007	0.037
KI67 (*n* = 174)	14.3 (±10.3, 11.6, 6.6–20.0)	14.9 (±10.3, 12.6, 7.6–20.8)	14.0 (±10.6, 12.0, 7.0–19.0)	8.46 (±8.5, 6.4, 4.5–8.5)	0.028	0.947	0.021
CD3 (*n* = 172)	1306.4 (±1041.9, 1115.6, 478.2–1822.6)	1204.1 (±981.7, 1047.8, 465.5–1717.0)	1505.6 (±1251.0, 1278.7, 475.0–2102.8)	1968.6 (±752.3, 1907.1, 1628.0–2332.0)	0.016	0.524	0.015
Granzyme B (*n* = 179)	132.8 (±206.5, 61.4, 18.4–156.4)	89.7 (±103.8, 52.3, 15.7–134.8)	201.8 (±240.3, 104.4, 35.0–287.6)	407.0 (±499.3, 220.9, 57.8–542.9)	0.0006	0.043	0.003
CD68 (*n* = 176)	342.5 (±307.4, 261.3, 128.5–461.9)	345.2 (±295.6, 275.8, 136.6–457.6)	359.8 (±373.8, 208, 125.5–487.8)	245.1 (±265.5, 154.7, 22.5–384.6)	0.407	-	-
MPO (*n* = 188)	31.0 (±67.3, 9.9, 0.77–28.8)	24.5 (±45.4, 9.6, 0–28.3)	47.8 (±114.1, 10.0, 1.7–25.6)	66.0 (±108.8, 17.3, 4.5–75.0)	0.380	-	-
CD34 + stromal fibroblasts (*n* = 175)	4.56 (±3.49, 3.90, 1.92–5.82)	5.00 (±3.54, 4.28, 2.39–6.63)	3.59 (±3.25, 2.94, 1.31–4.59)	2.36 (±2.09, 1.48, 0.87–3.46)	0.001	0.036	0.009
Inos (*n* = 163)	381.2 (449.8, 256.9, 131.0–452.3)	409.0 (±486.6, 268.6, 159.6–453.9)	320.9 (±303.6, 215.8, 94.8–460.7)	222.7 (±214.0, 111.2, 65.4–394.4)	0.103	-	-
IL10 (*n* = 147)	195.0 (±207.3, 122.6, 39.7–296.0)	184.5 (±191.9, 119.5, 48.2–235.5)	184.6 (±268.7, 47.2, 26.2–296.0)	339.1 (±183.6, 336.0, 201.0–435.0)	0.0195	0.466	0.044
TNFa (*n* = 148)	191.1 (±251.8, 109.9, 42.5–225.8)	163.4 (±225.2, 106.5, 41.6–181.3)	223.4 (±232.0, 162.7, 65.5–256.1)	416.7 (±416.5, 365.6, 23.4–692.8)	0.086	-	-
D240 (*n* = 194)	0.23 (±0.30, 0.13, 0.05–0.29)	0.25 (±0.32, 0.14, 0.05–0.33)	0.17 (±0.22, 0.12, 0.02–0.23)	0.13 (±0.12, 0.09, 0.06–0.14)	0.220	-	-
MHCII (*n* = 208)	756.9, (±612.5, 648.1, 318.6–1012.3)	752.8 (±627.4, 648.1, 290.2–1014.3)	819.9 (±594.6, 703.2, 349.0–1072.7)	640.9 (±492.5, 480.9, 360.4–912.8)	0.593	-	-
Mast cells (*n* = 185)	179.6 (±129.4, 141.8, 96.0–238.9)	162.6 (±107.1, 131.6, 89.1–208.9)	269.6 (±188.3, 234.3, 133–402)	147.6 (±92.3, 130.0, 88.0–205)	0.004	0.003	0.985

Data are presented by mean (±SD, median, p25 and p75). * Comparisons were performed using Kruskal–Wallis non-parametric tests; ** with post-hoc analysis for the comparisons between cardioselective beta-blocker (BB) versus no BB and non-cardioselective BB versus no BB (Nemenyi test) when global *p*-value was <0.05. Data are presented by their mean (±SD, median, p25 and p75).

**Table 4 cancers-12-01094-t004:** Survival analyses data.

Conditions	Univariate Analysis			Multivariate Analysis		
Outcome Assessed	Hazard Ratio	95%CI	*p*-Value	Hazard Ratio	95%CI	*p*-Value
Progression free survival *						
Use of beta-blockers						
Never (n = 205)	1.00	-	0.011	1.00	-	0.086
Before melanoma diagnosis (n = 57)	0.34	0.15–0.79	0.012	0.40	0.16–0.97	0.042
After melanoma diagnosis (n = 24)	0.38	0.14–1.07	0.067	0.59	0.21–1.65	0.317
Breslow thickness index	1.43	1.31–1.55	<0.001	1.30	1.18–1.44	<0.001
Age in categories						
<60	1.00	-		1.00	-	0.084
≥60	0.84	0.52–1.35	0.467	0.65	0.40–1.06	
Ulceration	7.07	4.12–12.13	<0.001	3.11	1.51–6.41	0.002
Melanoma related survival						
Use of beta-blockers						
Never (*n* = 205)	1.00	-	0.030	1.00	-	0.147
Before melanoma diagnosis (*n* = 57)	0.18	0.04–0.77	0.020	0.25	0.06–1.13	0.071
After melanoma diagnosis (*n* = 24)	0.41	0.12–1.41	0.159	0.58	0.18–1.92	0.372
Breslow thickness index	1.31	1.21–1.42	<0.001	1.24	1.08–1.41	0.002
Age in categories						
<60	1.00	-	0.049	1.00	-	0.088
≥60	0.55	0.30–0.99		0.56	0.29–1.09	
Ulceration	4.80	2.45–9.39	<0.001	1.83	0.50–6.66	0.359
Overall survival						
Use of beta-blockers						
Never (*n* = 205)	1.00	-	0.056	1.00	-	0.094
Before melanoma diagnosis (*n* = 57)	1.21	0.72–2.03	0.468	0.93	0.54–1.60	0.782
After melanoma diagnosis (*n* = 24)	0.19	0.05–0.81	0.025	0.20	0.05–0.86	0.030
Breslow thickness index	1.25	1.18–1.33	<0.001	1.23	1.13–1.33	<0.001
Age in years	1.04	1.03–1.06	<0.001	1.04	1.03–1.06	<0.001
Ulceration	3.75	2.24–6.28	<0.001	1.20	0.62–2.34	0.587

* Progression in melanoma disease or death due to melanoma were both considered as an event of disease extension.

**Table 5 cancers-12-01094-t005:** Distribution of the exposure to beta-blockers by molecule.

Beta-Blocker Molecule, *n* (%)	
Cardioselective beta-blocker before melanoma diagnosis	
atenolol	10 (23.8)
bisoprolol	5 (11.9)
labetalol	1 (2.4)
metoprolol	24 (57.1)
nebivolol	2 (4.8)
Wide spectrum beta-blocker before melanoma diagnosis	
carteolol ^a^	2 (13.3)
carvedilol	1 (6.7)
propranolol	4 (26.7)
sotalol	1 (6.7)
timolol ^a^	7 (46.7)
Cardioselective beta-blocker after melanoma diagnosis	
atenolol	4 (19)
bisoprolol	3 (14.3)
cotenolol	1 (4.8)
metoprolol	12 (57.1)
nebivolol	1 (4.8)
Wide spectrum beta-blocker after melanoma diagnosis	
propranolol	1 (33.3)
carvedilol	2 (66.7)

^a^ Eye drop formulation.

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
