# Peer review of "Effects of Beta-Blockers on Melanoma Microenvironment and Disease Survival in Human"

_cancers, 2020, doi:10.3390/cancers12051094_

Round 1

Reviewer 1 Report

My comments are the following.

1) The authors defined three groups of patients: 230 in the control group, 41 in the cardioselective beta-blockers group and 15 in the wide-spectrum beta-blockers group. The three groups have significant difference on age. In the multivariate survival analysis, the model was adjusted for age using a categorical variable (<60 vs. >60 years). It may be important to also test age as a continuous variable and be sure of the independent value of beta-blockers in this model.

2) Medical treatment (BRAFi, MEKi, immune checkpoint inhibitors) may considerably change the prognostic of patients with melanoma and should be take into account in the survival analysis. An adjustment to the medical strategy is of utmost importance to conclude of the protective effect of beta-blockers.

3) Definition of cardioselective and wide spectrum beta-blockers should be defined with the name of drugs in each subgroups in the method section.

Reviewer 2 Report

This is a retrospective study of primary melanoma biopsies and patient records. The article is quite well written, although some language polish might be good. The role of nor-adrenergic signalling in cancer progression, cellular stress responses and immune response is highly relevant, and this study adds to the increasing evidence for its importance. Many relevant markers have been investigated in a high number of samples. The current study provides associations, correlations and observations that suggests that nor-adrenergic signalling might be a target for therapeutic interventions. These are highly important observations. Still, mechanistic studies and prospective intervention studies are necessary. The article should be of interest to the Journal’s readers pending some revision.

Major:

  • M&M 4.1: This is a retrospective study, not prospective.
  • M&M: Beta blocker use. The authors need to describe how they obtained information on beta blocker use (i.e duration, indication, compliance, registries etc.), comorbidities. What procedures were used to avoid biases. What about obesity and smoking?
  • M&M 4.7: Some critical discussion on reproducibility and robustness of the quantification methods would be appreciated.
  • There was a significant impact on survival of age, at least in the univariate analyses. Age was also closely associated with beta blocker use. This potential bias is mentioned, but needs to be discussed in more detail.
  •  

Minor:

  • The title sounds strange, why “in human”?
  • There are more literature on melanoma and beta blockers that is highly relevant here: Lemeshow et al 2011
  • The number of patients using non-selective beta blockers is very low and includes eye drops. The authors should be very careful in concluding from this group.

Round 2

Reviewer 1 Report

The authors have added my different requests and have detailed appropriately bias regarding the survival analysis.